# *PUPAID*: A R + ImageJ pipeline for thorough and semi-automated processing and analysis of multi-channel immunofluorescence data

**Paul Régnier**[1,2]*, **Camille Montardi**[3○], **Anna Maciejewski-Duval**[1,2○], **Cindy Marques**[1,2,4,5], **David Saadoun**[1,2,4,5]

**1** Immunology-Immunopathology-Immunotherapy (i3) Laboratory, INSERM UMR-S 959, Sorbonne Université, Paris, France, **2** Biotherapy Unit (CIC-BTi), Inflammation-Immunopathology-Biotherapy Department (DHU i2B), Groupe Hospitalier Pitié-Salpêtrière, Assistance Publique-Hôpitaux de Paris (AP-HP), Paris, France, **3** Département de Médecine Interne, Hôpital Ambroise Paré, Assistance Publique-Hôpitaux de Paris (AP-HP), Université Paris Saclay, Boulogne-Billancourt, France, **4** Département de Médecine Interne et Immunologie Clinique, Groupe Hospitalier Pitié-Salpêtrière, Assistance Publique-Hôpitaux de Paris (AP-HP), Sorbonne Université, Paris, France, **5** Centre National de Référence Maladies Autoimmunes Systémiques Rares, Centre National de Référence Maladies Autoinflammatoires et Amylose Inflammatoire, Inflammation-Immunopathology-Biotherapy Department (DMU 3iD), Groupe Hospitalier Pitié-Salpêtrière, Assistance Publique-Hôpitaux de Paris (AP-HP), Sorbonne Université, Paris, France

○ These authors contributed equally to this work.
* paul.regnier@aphp.fr, paul.regnier@sorbonne-universite.fr

**Data Availability Statement:** PUPAID code is fully available as a GPLv3-licensed R package accessible from a dedicated GitHub repository

## Abstract

*PUPAID* is a workflow written in R + ImageJ languages which is dedicated to the semi-automated processing and analysis of multi-channel immunofluorescence data. The workflow is designed to extract fluorescence signals within automatically-segmented cells, defined here as Areas of Interest (AOI), on whole multi-layer slides (or eventually cropped sections of them), defined here as Regions of Interest (ROI), in a simple and understandable yet thorough manner. The included (but facultative) R Shiny-based interactive application makes *PUPAID* also suitable for scientists who are not fluent with R programming. Furthermore, we show that *PUPAID* identifies significantly more cells, especially in high-density regions, as compared to already published state-of-the-art methods such as *StarDist* or *Cellpose*. For extended possibilities and downstream compatibility, single cell information is exported as FCS files (the standardized file format for single cell-based cytometry data) in order to be openable using any third-party cytometry analysis software or any analysis workflow which takes FCS files as input.

## Introduction

Since the discoveries of optical microscopy during in the late-16th century by Hans and Zacharias Janssen and the fluorescence phenomenon by Sir John Frederick William Herschel in the mid-19th century [1], scientists made considerable improvements in the fluorescence microscopy field. In parallel, the emergence and development of electronic devices capable of acquiring with more and more sensitivity fluorescence signals as well as processing them strongly

(https://github.com/PaulRegnier/PUPAID). However, due to its own size, the example dataset is not directly included within PUPAID package. Instead, it is fully accessible from the Data Station Life Sciences repository hosted by the Data Archiving and Networked Services (DANS) via the following URL: https://doi.org/10.17026/LS/7XQFAT. Noteworthily, a complete tutorial which features and describes the full step-by-step code applied on the example dataset is also available at https://paul-regnier.fr/tutoriel-pupaid/. To help users to better apprehend PUPAID, we also provide a YouTube video which shows the full course of the workflow applied on the example dataset: https://youtu.be/58Tm54OVP-g.

**Funding:** The author(s) received no specific funding for this work.

**Competing interests:** The authors have declared that no competing interests exist.

contributed to the astonishing expansion of the fluorescence microscopy and imaging fields [2–11]. Nowadays, the immunofluorescence technique is commonly and broadly used in a variety of research topics, and notably in biology, immunology and oncology. During the last decade, it gained even more potential, as new multiplexing immunofluorescence techniques allowing to stain up to 100 markers simultaneously were massively developed [12–15].

Unfortunately, these gradual but major improvements also came with an important drawback incarnated by the exponential increase of the complexity of the generated images and datasets, especially regarding the number of acquired markers, the images resolution but also the size of the analyzed cohorts [12, 14, 15]. Among the different software intended to decorticate this impressive amount of data, commercially available ones rank among the most powerful analysis tools (such as InForm from Akoya Biosciences, HALO from Indica Labs or Visiopharm from the eponym brand). On the contrary, open-source and free counterparts (such as ImageJ, QuPath, CellProfiler or Napari), which are technically able to offer equivalent analyses, frequently come with a steeper learning curve and the "highly recommended-to-almost mandatory" need to use macros or scripts (and thus programming) in order to switch from manual sample-by-sample image treatments and analyses to efficient and automated high-throughput pipelines and workflows capable to handle more samples and datasets with a higher complexity [12, 16–20].

Noteworthily, the previous paragraph becomes less and less true, thanks to the appearance of a plethora of new analysis methods [12, 16, 17, 20–22], including high-level deep-learning- and neural network-based tools, such as *CellSighter* [23] and *CellSpotter* [24] which allow to segment cells, extract fluorescence information as well as cluster and classify cells according to their phenotypes and to *a priori* known references. Additionally, the vast majority of the available methods only seems to propose a given kind of analysis and almost never include the appropriate pre-processing steps for the raw images. Moreover, as time goes by, such innovative analysis methods require from users a continuously growing bioinformatic background, for instance related to the generation, tuning and use of models. Apart from the difficulty this can represent for the majority of scientists who are not fluent in programming, the training and tuning of models is often very time-consuming as well as computing intensive, thus making these steps restricted to GPUs and/or to high-end computing nodes.

Here, we propose a R package named *PUPAID* (*Pipeline for Unleashed Processing and Analysis of Immunofluorescence Data*) which is a complete processing and analysis workflow designed for sequential same-slide multiplex immunofluorescence data: through either its command line- or interactive R Shiny application-based modes, it is intended to transform image-embedded data to single cell-based information in a manner that is as unsupervised as possible with limited human intervention. What first sets *PUPAID* apart is its significantly better performance for cell segmentation and retrieval as compared to other already published cell segmentation methods like *StarDist* [25, 26] and *Cellpose* [27], notably in tissue areas with high cell confluence. With its most balanced σ parameters, *PUPAID* is overall (all measured metrics averaged) 32.11% and 40.8% more efficient than *StarDist* and *Cellpose*, respectively, to identify cells in highly-infiltrated regions, whereas it is 10.51% less efficient than *StarDist* but 16.09% more efficient than *Cellpose* in lowly-infiltrated regions. Second, the extracted information, such as cell coordinates, shape descriptors and fluorescence levels, are ultimately integrated in the form of all-in-one FCS files, which are thus ready to be analyzed through already adapted and experienced tools and software. Consequently, researchers are still free to analyze their results in the way they want to. We believe that *PUPAID* will help to standardize and improve both quality and speed of analyses which use multiplex immunofluorescence datasets and thus greatly ease the subsequent results production and interpretation by researchers and scientists.

## Material and methods

### Example dataset for learning and test purposes

For learning and test purposes, *PUPAID* includes a dataset composed of 17 tiles (TIFF format) which contain demultiplexed signals of an 8-plex immunofluorescence experiment performed using Opal technology from Akoya Biosciences, following their standard staining protocol. The staining was performed on a 5μm thick slide coming from FFPE-embedded salivary gland biopsy taken from a patient with Sjögren's syndrome and subsequently acquired with a Vectra Polaris scanner. The patient involved in the test dataset indeed gave informed consent for biopsies and associated clinical research. The staining panel (primarily designed to target and study the whole B and T cells compartments as well as several other features) was constructed as following: DAPI, anti-IL-21 (revealed with Opal 480), anti-CD21 (revealed with Opal 520), anti-TNFα (revealed with Opal 540), anti-CD4 (revealed with Opal 570), anti-IFNγ (revealed with Opal 620), anti-CD27 (revealed with Opal 650) and anti-CD20 (revealed with Opal 690). After acquisition, each of the generated tiles was demultiplexed using the InForm tool coming with the Vectra Polaris acquisition software. Due to its size, the example dataset is not directly included within *PUPAID* R package. Instead, it is fully accessible from the Data Station Life Sciences repository hosted by the Data Archiving and Networked Services (DANS) *via* the following URL: https://doi.org/10.17026/LS/7XQFAT. Of note, the test dataset does not allow to fairly generate biologically relevant results alone, as it represents only 1 acquisition from 1 patient, without any control tissue. It is rather intended to present the possibilities of *PUPAID* and show how to use it.

### Benchmarking of the different cell segmentation approaches

Several sections of this manuscript describe, quantify and evaluate the efficiency of different cell segmentation methods. This could either involve the comparison of *PUPAID* method performance with or without the use of local contrast enhancement (CLAHE) algorithm before cell segmentation or the comparison of the manual cell segmentation to *PUPAID*, *StarDist* and *Cellpose* methods. Regardless of the objective, we applied each corresponding method independently on 10 distinct crops of dimensions 200x200 pixels coming from the example dataset described above in the **Example dataset for learning and test purposes** subsection of the **Material and methods** section: 5 coming from high-density (HD) regions and 5 coming from low-density (LD) regions. From the subsequent cell segmentation results, we extracted the respective binary masks using ImageJ. Then, we used the ImageJ plugin *MiC* (accessible here: https://github.com/MultimodalImagingCenter/MiC) to compute intermediate metrics such as true positives (TP), false positives (FP) and false negatives (FN) indices, as well as 4 different final metrics (*Precision*, defined as $\frac{TP}{TP+FP}$, *Recall/sensitivity* defined as $\frac{TP}{TP+FN}$, *Jaccard* defined as $\frac{TP}{TP+FP+FN}$ and *F-measure* defined as $\frac{2TP}{2TP+FP+FN}$) for each Intersection over Union (IoU) threshold ranging between 0 and 1 with a step of 0.05. This standard benchmarking method allows to generate curves (with the x-axis showing the different IoU thresholds and the y-axis showing any metric of interest) that we can thoroughly compare between the methods/parameters used. Of note, we used the dedicated *StarDist* ImageJ plugin and the Python implementation with GUI of *Cellpose* in their latest versions and with their respective default parameters. For *StarDist*, the *Versatile (fluorescent nuclei)* model was used, and for *Cellpose V2*, the mean cell diameter was automatically determined and used within the *nuclei* model. Regarding *PUPAID* method, the initial *sigmaLow* value used for the related Difference of Gaussians (DoG) analyses was defined as 2.13, which is resulting from the $r = \sigma\sqrt{2\log(255)} - 1$ relation with an estimated mean cell radius ($r$) of 7.5μm, as measured with ImageJ. We then computed our

different *sigmaHigh* values as the *sigmaLow* value multiplied by different factors: 1.1, 1.2, 1.4, 1.5, 2, 5 and 10, which composed the different sets of σ ratios further described in this manuscript.

## Results and discussion

### Installation and dependencies

*PUPAID* relies on the following R packages in order to be installed and perform its routine normally: *devtools*, *foreach*, *flowCore*, *data.table*, *Biobase*, *methods*, *writexl*, *shiny* and *shinyFiles*, which are all accessible *via* either CRAN or Bioconductor repositories. These should normally be automatically downloaded during installation, except for the *devtools* package which must be manually installed with the following command: *install.packages("devtools")* then loaded with the following one: *library("devtools")*. After installation and loading of the *devtools* package, users only have to run the following command to install *PUPAID*: *devtools::install_github("PaulRegnier/PUPAID")*. When the installation is finished, *PUPAID* can be loaded with the following command: *library("PUPAID")*.

*PUPAID* also relies on the presence of the ImageJ software (or its improved version, Fiji), which can be respectively downloaded on the following websites: https://imagej.net/ij/download.html and https://imagej.net/software/fiji/downloads.

### General overview of *PUPAID*

*PUPAID* is designed as a tool to directly pre-process, process and analyze image files coming from sequential same-slide multiplex immunofluorescence experiments. It basically takes as input TIFF images of ROI that can be as large as the file format allows. At the moment of the publication of this manuscript, we did not implement methods to process immunofluorescence multiplexes generated with adjacent slides, although this could be implemented in the future if users show interest in such feature. The first objective of *PUPAID* is to process the fluorescence signals using background subtraction coupled to flatfield correction using the *rolling ball* algorithm [28] and by locally enhancing the contrast using CLAHE algorithm [29]. The second objective is to determine cell contouring using the well-known Difference of Gaussians (DoG) method [30–33], which is classically applied on intranuclear (using for instance 4′,6-diamidino-2-phenylindole, DAPI) or cell membrane staining (using for instance CellBrite Cytoplasmic Membrane Dyes from Biotium or CellMask Plasma Membrane Stains from ThermoFisher Scientific). Afterwards, the fluorescence of each remaining channel is computed for each segmented cell and exported in TSV- and XLSX-formatted tables, as well as other information such as the spatial coordinates of each cell and several descriptors of their shape (area, circularity, roundness and solidity). In parallel, the processed images are exported for each channel (either individually or in a single stack). Finally, *PUPAID* merges the exported TSV tables into a single FCS file per analyzed ROI. These FCS files are then ready to be opened and analyzed through any third-party cytometry analysis software, such as FlowJo (BD Life Sciences), Kaluza (Beckman Coulter), FCS Express (De Novo Software), BD FACSDiva (BD Biosciences), CytExpert (Beckman Coulter) and so on, or even by dedicated analysis pipelines like *PICAFlow* [34], *cytofkit* [35] or *CyTOF* [36] R packages. The **Fig 1** shows the detailed workflow as performed by *PUPAID*. Of note, apart from its standard command line-based implementation, *PUPAID* can also be fully used *via* an integrated R Shiny interactive application, which allows users who are not very familiar with programming to use *PUPAID* in an easier, more graphical and practical manner. Of note, a YouTube video showing the full course of the workflow is available to help future users to better apprehend *PUPAID*: https://youtu.be/58Tm54OVP-g.

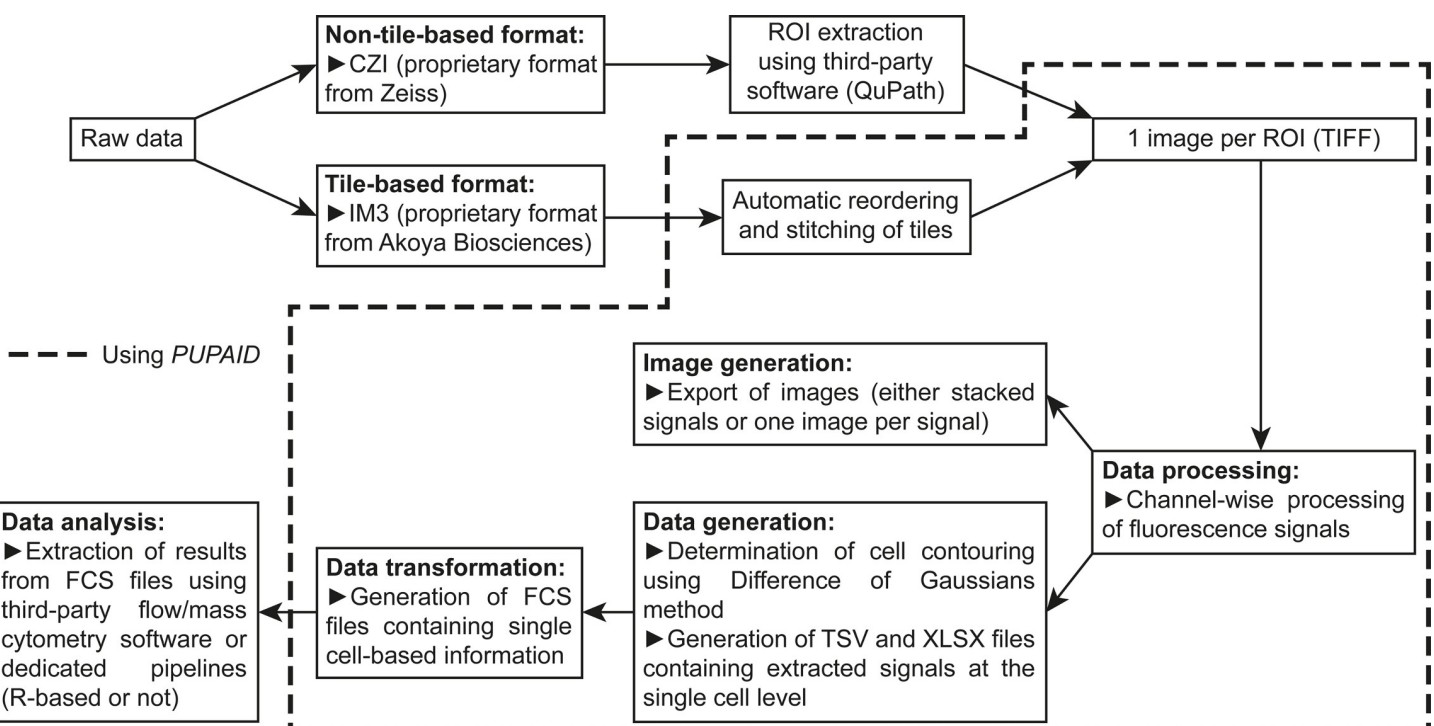

**Fig 1. Overview of *PUPAID* workflow.** *PUPAID* is able to process raw immunofluorescence data, extract cell-based information and convert them into TSV- and XLSX-formatted tables as well as into cytometry FCS files for easy and thorough analysis.

## Pre-processing of raw data

According to the raw data type, different pre-processing steps may be needed before the actual analysis part. If the acquired data is in a non-tile-based format, such as CZI (Zeiss), one should first open these with third-party software–like ZEN from Zeiss, or open-source ones such as QuPath (Windows and MacOS platforms), Fiji (all platforms) or *czifile* (Python-based)–in order to extract the desired ROI (as many as needed) in TIFF format (eventually LZW-compressed whenever possible to minimize disk usage). If the acquired data is rather in a tile-based format, such as IM3 (Akoya Biosciences), one should convert them to TIFF before using *PUPAID*. In the precise case of IM3 format, the demultiplexing feature included in the InForm analysis software from Akoya Biosciences already converts the tiles from IM3 to TIFF format during demultiplexing. Please note that this first step is indeed dependent on the acquisition platform used and may vary accordingly. In summary, *PUPAID* requires as input TIFF-formatted multi-channel images, either for a full ROI or for a batch of tiles. Importantly, *PUPAID* is able, without any human intervention, to automatically reorder and stitch a batch of tiles to regenerate one or several full ROI (depending on the contiguity of the provided tiles). *PUPAID* also proposes to generate *black tiles* (that is to say devoid of any fluorescence signal) which will eventually be used to reconstitute rectangular ROI with missing tiles. Noteworthily, at the moment, these features were only tested with tiles coming from demultiplexed IM3 raw data acquired using a Vectra Polaris scanner from Akoya Biosciences. That said, *PUPAID* could still be updated in the future to take other file formats as input if needed.

## Processing of ROI

In order to process the signals contained in each desired ROI, *PUPAID* separates each channel from the global stack and asks the user to manually give them a name for further identification. This renaming step is essential to correctly choose which channel should be used for the subsequent DoG algorithm application and which ones will be measured within the segmented cells. Usually, as DAPI is very commonly used in multi-channel immunofluorescence experiments for nuclei staining, this will likely represent the best possible choice. Nevertheless, this DoG method may also be used on other signals like cell membrane staining with dyes such as CellBrite Cytoplasmic Membrane Dyes from Biotium or CellMask Plasma Membrane Stains from ThermoFisher Scientific for instance. Please note that at the moment, *PUPAID* has only been tested with DAPI, but could eventually be updated to better integrate the treatment of other staining. After the selection of the desired channel, *PUPAID* will process it by subtracting the background and correcting for uneven illumination using the *rolling ball* algorithm [28] then by locally enhancing the contrast using CLAHE algorithm [29], and finally by applying the DoG algorithm for cell segmentation. By default, the CLAHE algorithm we implemented uses 30 pixels-sized blocks, 256 histogram bins, a maximum slope of 3 and uses the fast implementation of the filter. In all cases, users still have the possibility to edit these values if the default parameters do not produce satisfying enough results.

When the cell segmentation is finished, the user is asked to add a supplemental AOI in the dedicated mask within an area of the ROI which is devoid of any real fluorescence signal. This reference AOI will be used to compute the Corrected Total Cell Fluorescence (CTCF) for each channel of each segmented cell (see **Table 1** for the mathematical definition of CTCF). Afterwards, the remaining channels are also processed (including background subtraction and local contrast enhancing), and the fluorescence signals within each previously determined segmented cell as well as other information such as their spatial coordinates and several estimators of their shape (area, circularity, roundness and solidity, see **Table 1**) are computed and exported in TSV- and XLSX-formatted tables. In parallel, processed images of each channel (either one per channel or embedded within a single stack per ROI) are also exported as LZW-compressed TIFF images.

## Assessment of the CLAHE algorithm usefulness before *PUPAID*-mediated cell segmentation

First, we were interested to evaluate the usefulness of CLAHE algorithm before cell segmentation performed through *PUPAID*. To this end, we applied the methodology described in the **Benchmarking of the different cell segmentation approaches** subsection of the **Material and methods** section independently on each of the 10 crops previously described in the same subsection. The position of these crops on the example slide are shown in the **Fig 2A**. Red boxes indicate highly-infiltrated regions and yellow boxes indicate lowly-infiltrated regions. At first sight, the percentage of detected objects (relative to the ones obtained with manual segmentation) does not seem to decrease a lot in the condition where CLAHE was not used (blue) as compared to the full *PUPAID* protocol (black), although some statistically significant but slight decreases can be observed for LD regions (**Fig 2B**). On the contrary, when we look at 4 different metrics to compare the generated masks, we notice that *PUPAID* without CLAHE (dashed lines) globally leads to decreased *Precision*, *Recall/sensitivity*, *Jaccard* and *F-measure* as compared to their relative control including the CLAHE step (plain lines), both for HD (**Fig 2C**) and LD regions (**Fig 2D**), regardless of the σ ratio used. We also observe that at a fixed Intersection over Union (IoU) threshold of 0.4, these differences effectively reach significance (with less important values when CLAHE is not used), notably for the last 3 metrics between the

**Table 1. Metrics used or produced by *PUPAID*.** This table summarizes all the metrics which are used or produced through *PUPAID* workflow, as well as their full names, formulas and short descriptions. All metrics except CTCF are directly computed and exported using the *Measure* feature in ImageJ. CTCF for each parameter are computed afterwards *via* R during the generation of the merged FCS file using the formula presented in the table.

| Metric | Description |
|---|---|
| *Cell* | Unique identification number of an identified cell. |
| *Label* | Fluorescence parameter for which the associated measures were made. |
| *Area* | Area of an identified cell. |
| *Mean* | Average gray value for the fluorescence parameter specified by *Label* within an identified cell. For an identified cell, this is the sum of the gray values of all the pixels divided by the number of pixels. |
| *X* | X-axis location of the centroid of an identified cell. For an identified cell, the centroid is defined as the center point, which is the average of the X coordinates of all the pixels. |
| *Y* | Y-axis location of the centroid of an identified cell. For an identified cell, the centroid is defined as the center point, which is the average of the Y coordinates of all the pixels. |
| *Circ.* | Short for *Circularity*. Defined as $4\pi \; x \left(\frac{Area}{Perimeter^2}\right)$. Considered as a shape descriptor of an identified cell. A circularity value of 1 indicates a perfect circle. As the value approaches 0, it indicates an increasingly elongated polygon. |
| *IntDen* | Short for *Integrated Density*. For an identified cell, defined as *Area × Mean* for the fluorescence parameter specified by *Label*. |
| *RawIntDen* | Short for *Raw Integrated Density*. For an identified cell, defined as the sum of the values of the pixels for the fluorescence parameter specified by *Label*. |
| *AR* | Short for *Aspect Ratio*. Defined as $\frac{Major \; axis}{minor \; axis}$. Considered as a shape descriptor of an identified cell. |
| *Round* | Short for *Roundness*. Defined as $\frac{4 \; x \; Area}{\pi \; x \; (Major \; axis)^2}$. Considered as a shape descriptor of an identified cell. |
| *Solidity* | Defined as $\frac{Area}{Convex \; area}$. Considered as a shape descriptor of an identified cell. |
| *CTCF* | Short for *Corrected Total Cell Fluorescence*. Defined as *IntDen–(Area × Mean fluorescence of background)*. Represents the fluorescence level of a cell for the marker specified by *Label* once the background is subtracted. |

1:1.1 and 1:2 σ ratios (**Fig 2E**). In other words, even if it does not seem to have a great impact on the overall percentage of detected objects, the use of the CLAHE algorithm after background correction seems essential for *PUPAID* to increase the efficiency and the quality of the upcoming cell segmentation, and notably regarding the improved *Recall/sensitivity*, *Jaccard* and *F-measure* metrics as compared to the *PUPAID* version without CLAHE. In our opinion, this step should be performed on all images which are analyzed through *PUPAID*, as it allows to more precisely recover and identify cells within the images as compared to the version of the workflow where the CLAHE step is lacking.

## Benchmarking of either *PUPAID*-, manual- or other state-of-the-art methods-generated cell segmentation

Afterwards, we wanted to investigate how *PUPAID* performed as compared to other cell segmentation methods and how its performance can be optimized. The main goals of this benchmarking are 1) to assess how *PUPAID* performs in terms of cell segmentation as compared to either manual or other state-of-the-art methods (herein *StarDist* or *Cellpose*), and 2) to determine the best pair(s) of σ values, representing the intensity of image blurring during the DoG algorithm, which *in fine* lead to the best cell segmentation with *PUPAID* in the tissue of interest. These σ parameters are crucial in this process and should be determined with great care. Using the previously mentioned 200x200 pixels crops exposed in **Fig 2A** as well as the methodology described in the **Benchmarking of the different cell segmentation approaches** subsection of the **Material and methods** section, we compared *PUPAID* performance with different

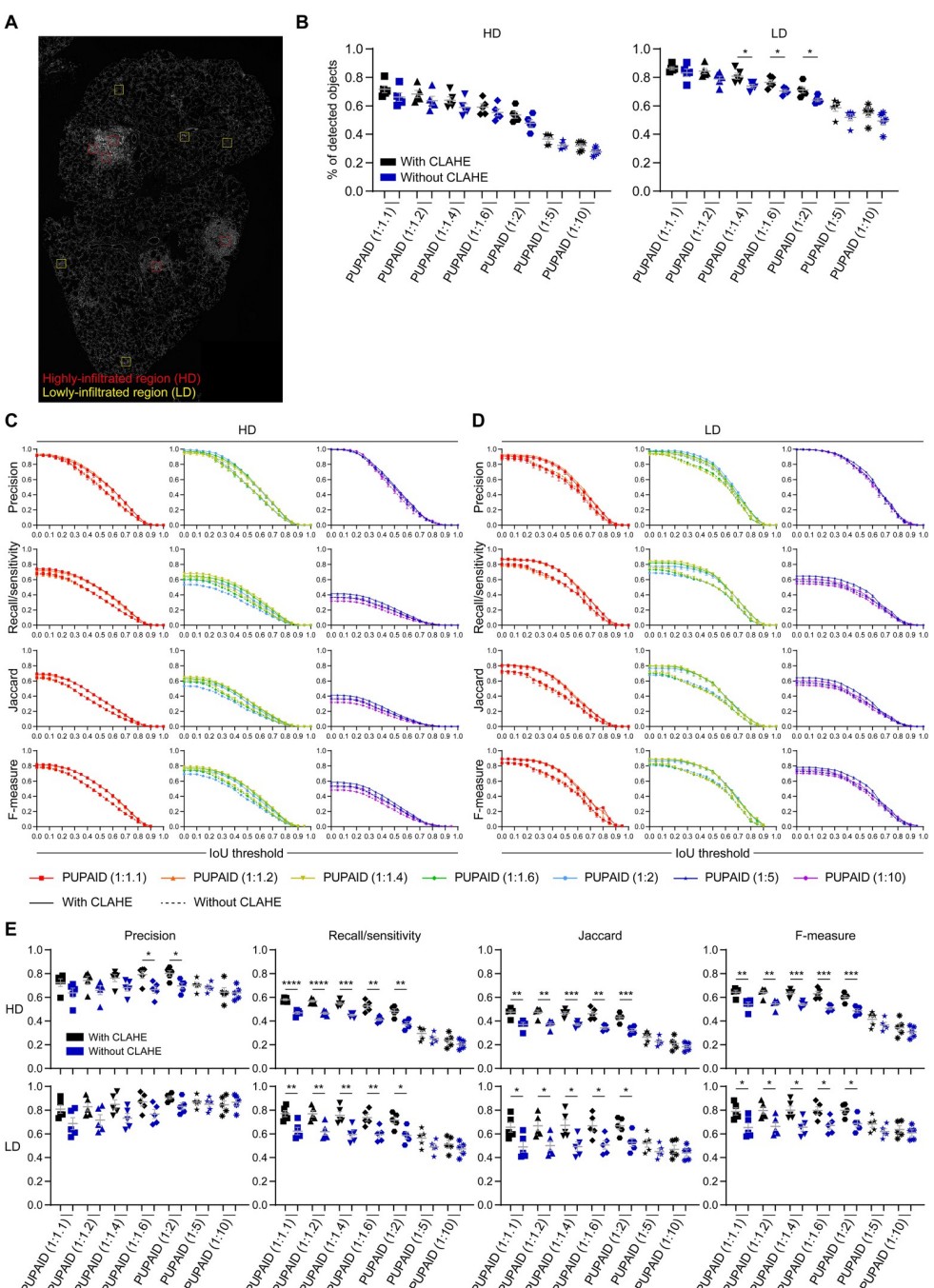

**Fig 2. Local contrast enhancing using CLAHE algorithm is necessary for optimized cell segmentation by *PUPAID*.**
**(A)** Stitched image from the test dataset which is overlaid with the 10 200x200 pixels crops we generated to evaluate
*PUPAID*'s performance. 5 of them are located in high-density regions (red squares) and the remaining 5 are located in
low-density regions (yellow squares). Then, we applied on these crops the methodology described in the
**Benchmarking of the different cell segmentation approaches** subsection of the **Material and Methods** section of the
manuscript. **(B)** Percentage of detected objects for each depicted method in HD (left) and LD (right) regions as
compared to the manual cell segmentation which represents 100% of the cells. **(C-D)** Graphs showing the Intersection
over Union (IoU) thresholds vs. the 4 quality metrics (*Precision*, *Recall/sensitivity*, *Jaccard* and *F-measure*) generated by
the *MiC* ImageJ plugin for the depicted methods in HD **(C)** and LD **(D)** regions. **(E)** Same data as previously shown in
**(C-D)** except that this subfigure focuses on an IoU threshold of 0.4 for HD (top) and LD (bottom) regions. Groups
were compared using Student's t-test and p-values were reported as follows: *: p-value < 0.05, **: p-value < 0.01, ***:
p-value < 0.001 and ****: p-value < 0.0001. Data are represented as mean±SEM.

σ pairs against manual segmentation made by eye, but also against *StarDist* and *Cellpose* already published methods. A representative cell segmentation example for 1 HD and 1 LD regions of the original image coming from the test dataset, as well as the obtained masks, is shown in **Fig 3A**.

At first sight (as seen in **Fig 3A**), we observe that *PUPAID* seems to perform better than *StarDist*, especially in the HD region, and roughly comparably to *Cellpose*. In the LD region, all methods seem to give similar results. To precisely quantify these first observations, we primarily measured the percentage of detected objects (**Fig 3B**) achieved for each method as compared to the reference (equal to 100% for manual cell segmentation), which clearly show that for HD regions, *PUPAID* method with σ ratios ranging between 1:1.1 and 1:2 leads to a marked and significant increase of the percentage of detected objects as compared to *StarDist* or *Cellpose*. For LD regions, only σ ratios ranging between 1:1.1 and 1:1.6 lead to non-significantly different percentages of detected objects as compared to *StarDist* (while *Cellpose* always performs worse than *StarDist* or *PUPAID*). We also notice that the σ ratios of 1:5 and 1:10 lead to poor results, which are comparable to both *StarDist* and *Cellpose* for HD regions, and comparable to *Cellpose* only for LD regions.

Then, we generated the curves presented in **Fig 3C** (left column for HD and right column for LD) which represent the variation of each quality metric (y-axis) over IoU thresholds (x-axis). For HD, we observe that *PUPAID* globally leads to decreased *Precision* values as compared to *StarDist*. Remarkably, we notice that *PUPAID* (for the σ ratios comprised between 1:1.1 and 1:2) systematically leads to an important increase of the *Recall/sensitivity*, *Jaccard* and *F-measure* metrics as compared to *StarDist* or *Cellpose*. For LD regions, it appears that the *Precision* is overall slightly decreased with *PUPAID* as compared to *StarDist* or *Cellpose*. Similarly, the 1:5 and 1:10 σ ratios for *PUPAID* performed very poorly regarding the *Recall/sensitivity*, *Jaccard* and *F-measure* metrics, reaching a level comparable to *Cellpose*. On the other hand, the values for these 3 last metrics are much closer to the ones obtained with *StarDist* when the other σ ratios (between 1:1.1 and 1:2) for *PUPAID* are used. More precisely, the 1:2 σ ratio performs globally less well than the 1:1.1, 1:1.2, 1:1.4 and 1:1.6 σ ratios, suggesting that these 4 last ones could represent the optimal σ values range for cell segmentation using DoG method on this precise tissue. On the contrary, these 4 metrics allow to determine that 1:5 and 1:10 σ ratios for *PUPAID* are the ones which clearly show the worst performance as compared to other methods/parameters (and notably *Cellpose*) and that they are indeed not optimal for DoG-based cell segmentation, at least in the context of the tested tissue.

The previous observations are further confirmed if we focus on an IoU threshold of 0.4 (**Fig 3D**). Indeed, we clearly see for HD regions that σ ratios comprised between 1:1.1 and 1:2 lead to significantly lower *Precision* values as compared to *StarDist*, but to similar ones as compared to *Cellpose*. Of note, the decrease in the *Precision* metric is not very important in terms of magnitude, with for instance a ≈11% decrease between *StarDist* (mean±SEM of 0.8894±0.0184) and *PUPAID* with a σ ratio of 1:1.6 (mean±SEM of 0.7910±0.03144). Strikingly, the same σ ratios range for *PUPAID* lead to significantly increased *Recall/sensitivity*, *Jaccard* and *F-measure* values as compared to both *StarDist* and *Cellpose*. For LD regions, we observe the same decrease of the *Precision* value for *PUPAID* as compared to *StarDist*. Of note, the other metrics are also affected and significantly decrease when *PUPAID* is used as compared to *StarDist*, regardless of the σ ratio used. These decreases are again rather limited, with for instance a ≈10%, ≈12%, ≈18% and ≈11% decrease between *StarDist* and *PUPAID* with a σ ratio of 1:1.6 for *Precision*, *Recall/sensitivity*, *Jaccard* and *F-measure*, respectively. Overall, in LD regions, it appears that *PUPAID* globally performs in a comparable manner to *Cellpose* regarding *Precision* but better than it regarding the other metrics. These results also suggest that optimal σ

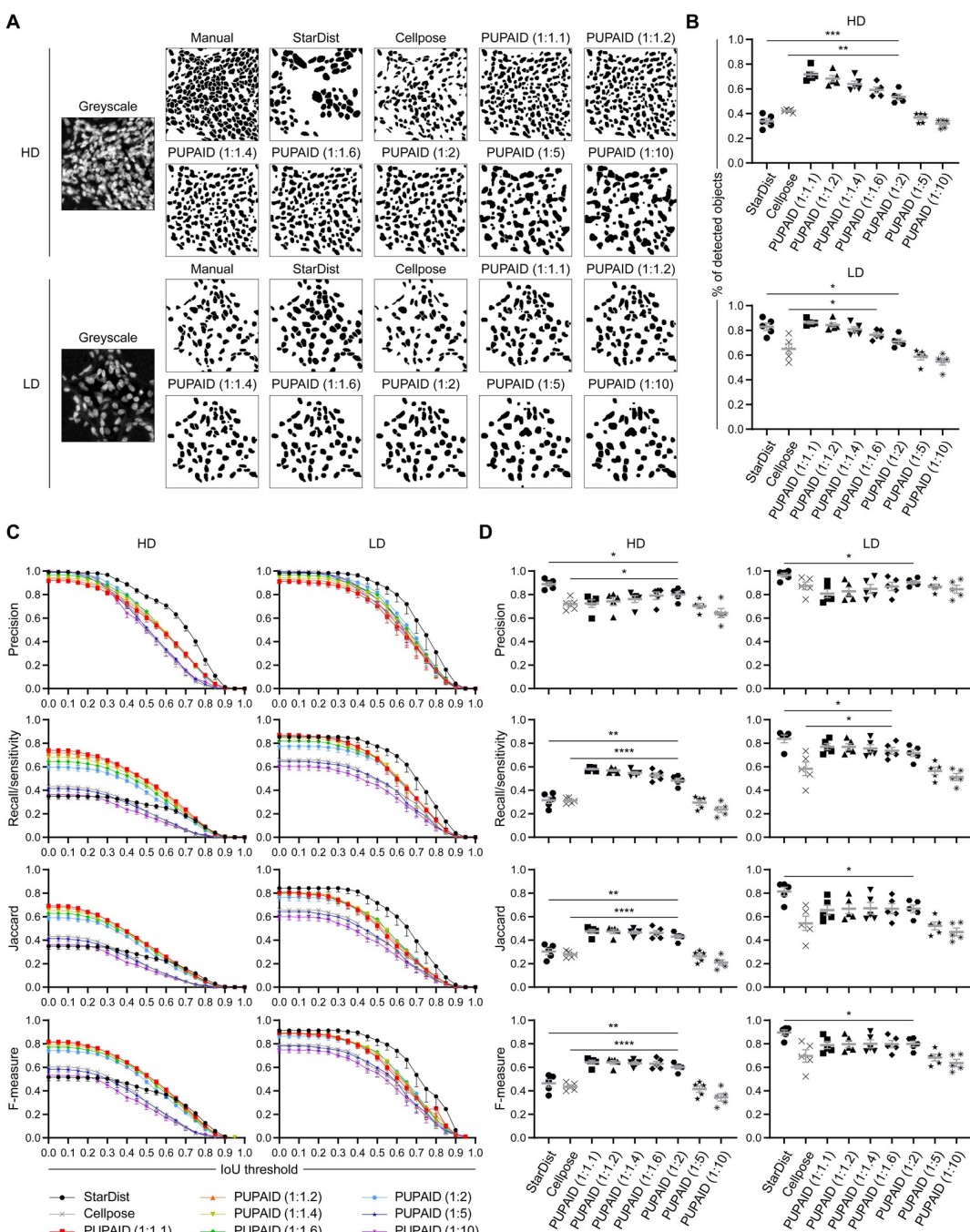

**Fig 3. Benchmarking of either _PUPAID_-, manual- or other state-of-the-art methods-generated cell segmentation. (A)** Original greyscale images and associated cell segmentation masks generated with the depicted methods from 1 representative high-density (HD) and 1 representative low-density (LD) crop coming from the test dataset. We extracted 10 distinct crops of dimensions 200x200 pixels from the test dataset: 5 coming from HD regions and 5 coming from LD, as respectively shown by the red and yellow squares in the **Fig 3A**. Then, we applied on these crops the methodology described in the **Benchmarking of the different cell segmentation approaches** subsection of the **Material and methods** section. **(B)** Percentage of detected objects for each depicted method in HD (top) and LD (bottom) regions as compared to the manual cell segmentation which represents 100% of the cells. **(C)** Graphs showing the Intersection over Union (IoU) thresholds vs. the 4 quality metrics (_Precision_, _Recall/sensitivity_, _Jaccard_ and _F-measure_) generated by the _MiC_ ImageJ plugin (accessible here: https://github.com/MultimodalImagingCenter/MiC) for the depicted methods in HD (left) and LD (right) regions. **(D)** Same data as previously shown in **(C)** except that this subfigure focuses on an IoU threshold of 0.4 for HD (left) and LD (right) regions. Groups were compared using Student's t-test and p-values were reported as follows: *: p-value < 0.05, **: p-value < 0.01, ***: p-value < 0.001 and ****: p-value < 0.0001. Data are represented as mean±SEM.

ratios for *PUPAID* cell segmentation range somewhere between 1:1.1 and 1:2, at least in the context of the tested tissue.

Overall, it appears that *PUPAID* performs similarly or even slightly better than *Cellpose* in LD regions (and comparably or slightly less well than *StarDist*), but seems to outperform these two methods in HD regions: despite a significant but limited decrease in the *Precision* metric, *PUPAID* leads to significantly higher *Recall/sensitivity*, *Jaccard* and *F-measure* metrics as well as percentages of detected objects as compared to *StarDist* or *Cellpose*. Additionally, we propose to consider the 1:1.4 σ ratio as the most balanced for cell segmentation using *PUPAID*, at least in this example dataset. The **Table 2** summarizes the average results obtained for each measurement (the percentage of detected objects as well as the 4 previously mentioned metrics) using the 1:1.4 σ ratio and an IoU threshold of 0.4.

## Integration of multiple TSV files to all-in-one FCS file

At the very end of the workflow, *PUPAID* offers the possibility to convert the previously exported TSV files to new XSLX files which are compatible with Microsoft Excel as well as to transform the exported TSV tables into a single FCS file. These TSV/XSLX files contain a lot of metrics about each of the segmented cells, which are all described in the **Table 1**. The subsequent FCS file transformation allows to make the visualization and final analysis process easier by letting users open these files with any cytometry analysis software of their choice, like FlowJo (BD Life Sciences), Kaluza (Beckman Coulter), FCS Express (De Novo Software), BD FACSDiva (BD Biosciences) or CytExpert (Beckman Coulter) for instance. As shown in the **Fig 4**, users can gate on cells of interest using either their shape descriptors (area, circularity, roundness and solidity), their actual spatial coordinates within the tissue or their phenotype and even combine all these approaches to mimic real cytometry analyses through personalized gating strategies. Importantly, the FCS-based results integration also allows users to analyze these files through dedicated bioinformatic analysis pipelines, such as *PICAFlow* [34], *cytofkit* [35] and *CyTOF* [36] R-based approaches, for instance. To illustrate, the top-left plot shows a gate which only selects the cells which express DAPI the most and present the highest areas. The cells which are very small or express DAPI very lowly are probably artifacts or poorly segmented cells and need to be discarded. The top-right plot shows the cell content of the gate

**Table 2. Summary of the efficacies of the compared cell segmentation methods.** The table presents the mean values independently for each measurement, which include the percentage of detected objects and the *Precision*, *Recall/sensitivity*, *Jaccard* and *F-measure* metrics generated by the *MiC* ImageJ plugin. The presented values were computed with an IoU threshold of 0.4 and with the 1:1.4 σ ratio for *PUPAID*. The efficacy of a given method is reported as a percentage between 0 and 1 which is relative to the manual cell segmentation.

| | Region | Efficacy of *StarDist* | Efficacy of *Cellpose* | Efficacy of *PUPAID* | % of evolution from *StarDist* to *PUPAID* | % of evolution from *Cellpose* to *PUPAID* |
|---|---|---|---|---|---|---|
| **% of detected objects** | HD | 0.3375 | 0.4223 | 0.6395 | +89.48% | +51.43% |
| | LD | 0.8292 | 0.6506 | 0.8088 | -2.46% | +24.32% |
| **Precision** | HD | 0.8894 | 0.7192 | 0.7616 | -14.37% | +5.9% |
| | LD | 0.9656 | 0.8738 | 0.8482 | -12.16% | -2.93% |
| **Recall/sensitivity** | HD | 0.3150 | 0.3118 | 0.5476 | +73.84% | +75.63% |
| | LD | 0.8376 | 0.5826 | 0.7568 | -9.65% | +29.9% |
| **Jaccard** | HD | 0.3036 | 0.2784 | 0.4664 | +53.62% | +67.53% |
| | LD | 0.8142 | 0.5436 | 0.6718 | -17.49% | +23.58% |
| **F-measure** | HD | 0.4636 | 0.4350 | 0.6354 | +37.06% | +46.07% |
| | LD | 0.8956 | 0.6964 | 0.7998 | -10.7% | +14.85% |
| **Average efficacy** | HD | 0.4618 | 0.4333 | 0.6101 | +32.11% | +40.8% |
| | LD | 0.8684 | 0.6694 | 0.7771 | -10.51% | +16.09% |

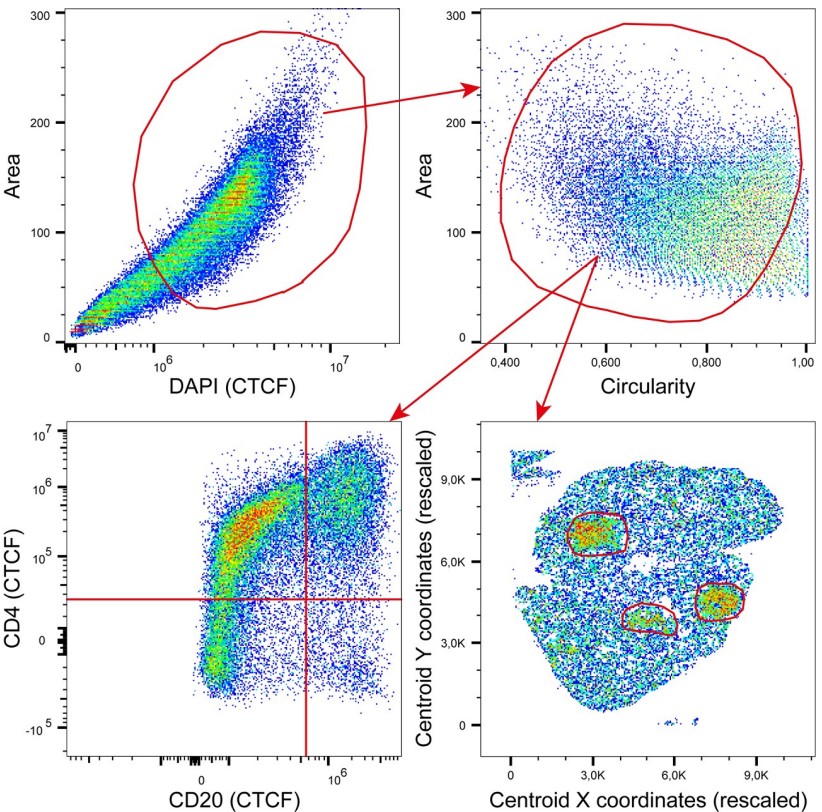

**Fig 4. Gating strategy-based analytic approach on the produced FCS files.** The final goal of *PUPAID* is to produce
FCS files which can be conveniently opened and analyzed in conventional flow/mass cytometry software such as BD
FACSDiva (BD Biosciences), FlowJo (BD Life Sciences), CytExpert (Beckman Coulter), CyTOF (Fluidigm), Kaluza
(Beckman Coulter), or FCSExpress (De Novo Software) or through dedicated R-written all-in-one analysis pipelines
such as *PICAFlow* [34], *cytofkit* [35] or *CyTOF* [36]. Each plot can represent a given dimension of the data generated
during *PUPAID* processing and analysis, and each dimension represents an aspect of each cell that was segmented
during the analysis, such as areas, X/Y coordinates, shape descriptors or fluorescence markers. Red circles indicate
examples of gates that can be drawn in order to extract the cells of interest.

displayed in the top-left plot. Here, we are looking at a shape descriptor (the circularity) and
mainly decide to discard the cells which are almost perfectly round (circularity close to 1),
which is unlikely to happen in real life. Then, the bottom plots show 2 different ways to analyze
the remaining cells. The bottom-left plot shows how they express CD20 and CD4, which is
very helpful to immediately distinguish B lymphocytes (CD20$^+$ CD4$^-$) and CD4$^+$ T lympho-
cytes (CD20$^-$ CD4$^+$). The bottom-right plot rather shows the X/Y coordinates corresponding
to each segmented cell, which allows to analyze the tissue in a spatial manner, for instance here
by gating on cell clusters defined in anatomic pathology as a "focus" in the salivary gland. One
could for instance compare these highly infiltrated areas with other ones in order to evaluate
the phenotype of the infiltrating cells and/or their specificity.

## Application of *PUPAID* to already published studies

To date, *PUPAID* workflow was already successfully applied in 2 published studies. First, we
used this workflow in a work published last year in the *Circulation Research* journal, aiming to
better study the pathophysiology of Giant Cell Arteritis (GCA), one of the most common large
vessel vasculitis in humans [37]. Using several complementary experimental techniques, we

demonstrated that CTLA-4 transcript and protein were significantly and specifically upregulated both in blood and aorta of GCA patients as compared to healthy donors (HD). Of note, we performed, in the **Fig 3** of the paper, 4-plex and 8-plex immunofluorescence experiments on GCA aorta and used *PUPAID* to process and analyze the generated raw data. We demonstrated 1) that infiltrating regulatory T cells (Tregs, characterized as CD4$^+$ FoxP3$^+$ cells) are less abundant in GCA vs. HD, 2) that CTLA-4 was upregulated in CD4$^+$ T cells (and more importantly in Tregs) from GCA compared to their HD counterparts, and 3) that infiltrating Tregs are much less activated/suppressive in GCA as compared to HD. Importantly, these results were indeed confirmed by other approaches, such as transcriptomics and flow cytometry.

We also successfully used *PUPAID* in another study, also published last year in the *Arthritis and Rheumatology* journal, which involved this time skin biopsies coming either from controls or patients suffering from Behçet's disease (BD) [38]. As presented in the **Fig 2** of the paper, we evidenced a marked recruitment of neutrophils (CD66b$^+$ cells) as well as their increased expression of PDE4 protein in BD patients as compared to controls. We confirmed these results with other techniques such as transcriptomics and flow cytometry. This work helped to highlight that neutrophils are central in the pathophysiology of BD and that they are overactivated and overexpress PDE4 in this disease. Our results also imply that PDE4 could therefore represent a potential new target for immune intervention in BD.

Finally, we also applied *PUPAID* in another dataset, which consisted of salivary gland biopsies coming from patients suffering from Sjögren's syndrome. Briefly, this disease is associated with an elevated risk (10–15 times more) to develop lymphoma. Using different subgroups of Sjögren patients, we identified cell populations among B and T cells compartments which are increasing through the lymphoproliferation and are massively expanded in the Sjögren patients who developed lymphoma. We confirmed these results and went further with other techniques such as flow cytometry and single cell RNA sequencing. Of note, at the time of the submission of *PUPAID*'s manuscript, this work on Sjögren's syndrome is still under finalization.

## Other model-based cell segmentation methods

Although we were not able to test them herein, it is important to mention that during the last years, new promising tools appeared, such as *CellSighter* [23] and *CellSpotter* [24], which can prove very useful to segment and identify cells on high-throughput immunofluorescence datasets. Nowadays, such new methods are essentially constructed as deep-learning- or neural networks-based methods, which therefore makes them model-based and consequently require users to manage the notion of model construction using training and annotated data. Such approaches, even if potentially powerful and very precise, are time-consuming regarding the generation and annotation of the training data, and often require the access to GPUs and/or to high-end computing clusters in order to perform the model construction in a reasonable amount of time. Plus, such model generation makes the mastering of programming almost mandatory. To overcome these issues, the use of generic or already-trained models (if available, which is not always the case) could potentially ensue the generation of suboptimal results because the model was not fine-tuned for the target dataset. This could favor the creation of artefactual signals potentially leading to erroneous biological conclusions. Furthermore, even these new methods continue to rely on DoG-based or derived algorithms for cell segmentation, like it is the case for *CellSpotter*, which essentially uses an approach comparable to *PUPAID*, except that the used thresholding algorithm is not the same (ISODATA [39] for *PUPAID* and Otsu [40] for *CellSpotter*). This is why we believe that the use of optimized

versions of simple but already well-proven and robust methods such as the DoG algorithm for cell segmentation probably still represents the best choice for the analysis of multiplex immunofluorescence data. That said, these new approaches are indeed very helpful and promising, in the sense that they rather focus their workflows towards clustering and classification of identified cells using a combination of their respective phenotypes and already known reference ones, which is a very interesting and biologically pertinent feature that is not proposed by *PUPAID* at the moment.

### Potential impact of *PUPAID* on the clinical and medical fields

Thanks to its performance, especially in high-density regions, we believe that *PUPAID* could prove useful in the clinical and medical fields. For instance, kidney biopsies are frequently taken from patients in a variety of contexts, such as anti-neutrophil cytoplasmic antibody (ANCA)-associated vasculitis, anti-glomerular basement membrane disease (Goodpasture syndrome) or systemic lupus erythematosus [41–50]. Immunofluorescence-based assays targeting several markers such as immunoglobulins (IgG, IgA, IgM), complement proteins (C1q, C3) or other antigens such as myeloperoxidase (MPO), proteinase-3 (PR3) or fibrin/fibrinogen are already used for diagnostics and/or to better separate patients into different medical subentities. Importantly, the emergence of high-throughput techniques such as imaging mass cytometry led to a drastic improvement of the phenotyping process with up to 50 simultaneous targets instead of barely 10 [51]. Noteworthily, this first cited set of markers is also of interest in skin biopsies taken in other (but sometimes close) contexts like cutaneous vasculitis, pemphigoid and pemphigus diseases, systemic lupus erythematosus as well as other dermatoses and connective tissue disorders [52–60]. Another interesting example is incarnated by the inflammatory bowel diseases field which uses immunofluorescence since decades to study intestinal biopsies, initially with anti-IgG only and now with more than 30-plex panels *via* imaging mass cytometry [61, 62]. All of these contexts, as well as others not presented here (notably in the whole oncology field), could potentially profit from automated and more resilient analyses methods of multiplex immunofluorescence data in tissues like the one we expose herein with *PUPAID*. The broader use of multiplex immunofluorescence panels or any other high-throughput techniques in medicine and/or research, coupled to thorough and automated analysis methods could therefore dramatically improve the understanding of the diseases' pathophysiology, accelerate the discovery of biomarkers, and also help to further refine the classification of patients (which can still be tricky at the moment).

### Conclusions

To summarize, *PUPAID* is a R package which helps to automatize and systematize the processing and analysis of sequential same-slide multiplex immunofluorescence experiments. Its final objective is to bring to immunofluorescence-based datasets a clear, robust and reproducible processing and analysis workflow. We strongly believe that *PUPAID* could dramatically help scientists and researchers to extract more information from their immunofluorescence-based datasets with an increased statistical power and in a less supervised manner. Of note, its performance regarding cell segmentation as compared to manual-, *StarDist-* and *Cellpose-*generated cell segmentation, especially in high-density regions, clearly makes *PUPAID* a method of choice for the processing and analysis of sequential same-slide multiplex immunofluorescence data. Indeed, *PUPAID* could still be updated in the future to add more features, such as a broader compatibility with other kind of datasets (which was not fully tested yet in this version of the software) or even the automatic alignment of adjacent tissue sections.

## Author Contributions

**Conceptualization:** Paul Régnier.

**Data curation:** Paul Régnier, Anna Maciejewski-Duval.

**Formal analysis:** Paul Régnier, Camille Montardi, Anna Maciejewski-Duval.

**Investigation:** Paul Régnier, Camille Montardi, Anna Maciejewski-Duval.

**Methodology:** Paul Régnier, Anna Maciejewski-Duval.

**Project administration:** Anna Maciejewski-Duval, David Saadoun.

**Resources:** Paul Régnier, Anna Maciejewski-Duval, Cindy Marques.

**Software:** Paul Régnier.

**Supervision:** Anna Maciejewski-Duval, David Saadoun.

**Validation:** Paul Régnier.

**Visualization:** Paul Régnier.

**Writing – original draft:** Paul Régnier, Cindy Marques, David Saadoun.

**Writing – review & editing:** Paul Régnier.

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
