## [Decision Letter · Decision Letter 0]

14 Jun 2024

PONE-D-24-13313PUPAID: a R + ImageJ pipeline for thorough and semi-automated processing and analysis of multi-channel immunofluorescence dataPLOS ONE

Dear Dr. Régnier,

Thank you for submitting your manuscript to PLOS ONE. After careful consideration, we feel that it has merit but does not fully meet PLOS ONE’s publication criteria as it currently stands. Therefore, we invite you to submit a revised version of the manuscript that addresses the points raised during the review process.

**The revisions were positive, highlighting the manuscript's potential impact in research fields. However, while your work presents groundbreaking potential, its real advantages seem to be primarily in high-density regions. Given the clinical expertise among the authors, a detailed medical discussion would be highly beneficial.**

**From a reviewer's perspective, providing or at least describing the testing dataset is essential for comprehensive evaluation. Additionally, the manuscript lacks a thorough explanation of the benchmarking, which appears somewhat confusing. Improved communication on the quantitative methods used for benchmarking would significantly enhance the validation of PUPAID.**

We look forward to receiving your revised manuscript.

Kind regards,

Luca Pesce, Ph.D.

Academic Editor

PLOS ONE

Journal Requirements:

Reviewers' comments:

Reviewer's Responses to Questions

**Comments to the Author**

1. Is the manuscript technically sound, and do the data support the conclusions?

Reviewer #1: Partly

Reviewer #2: Yes

2. Has the statistical analysis been performed appropriately and rigorously? 

Reviewer #1: No

Reviewer #2: Yes

3. Have the authors made all data underlying the findings in their manuscript fully available?

Reviewer #1: No

Reviewer #2: Yes

4. Is the manuscript presented in an intelligible fashion and written in standard English?

Reviewer #1: No

Reviewer #2: Yes

5. Review Comments to the Author

**Reviewer #1:** Thank you for your contribution with the development of PUPAID. While your work presents groundbreaking potential, according to this version of the manuscript the real advantages appear to be primarily in high-density regions. Given the clinical expertise available among the authors, it would be highly beneficial to leverage this by providing detailed medical discussion.

From a reviewing point of view it would help to be provided with a testing dataset, or at least to describe it, for comprehensive evaluation. Additionally, the manuscript lacks a thorough explanation of the benchmarking which appears a bit confusing. I think venturing in many comparisons, different in nature, is not simple to deal with. Improved communication on the quantitative methods used for benchmarking would significantly enhance the validation of PUPAID.

I also think the paper lacks demonstration of user-friendliness, with the language used resulting in an additional barrier. Including a figure or, preferably, a video of the interface would greatly enhance understanding and usability for readers.

Lastly, the manuscript's language is quite self-referential and would benefit from a more reader-centric approach, ensuring clarity and accessibility for a broader audience. I suggest giving more scientific context and sometimes guiding the reader via bullet points.

Minor changes:

Abstract

Line 58: Would be nice for reviewers to have test files to run the code

Introduction

Line 82-83: would appreciate if the author could elaborate on what is meant by “exponential increase of the complexity”. Example: what is the role of noise and how is it evolving through time. Also it is not very clear what is an “early lack of open-source methods”

Line 88: The learning curve of coding within publicly available tools can be steep, conversely I would not define programs like Image J cumbersome to master.

Line 94: would be interesting if the author could elaborate on the relevance of Doc Parra in assessing methodology performances. Additionally, please provide further evidence of the status quo claimed here.

Lines 110-112: quite a statement about performance, please provide quantitative insights.

Dependencies

Lines 124-126: could the author stress further the decision of relying on R rather than python packages. Are R packages flexible in terms of being integrated on cloud and how suitable are they for integration with AI workflows?

Pre processing of raw data

Line 161: there are current efforts from the github community to manage such exotic formats https://github.com/cgohlke/czifile feel free to integrate in your script

Line 169: Is the “big idea” the innovation?

Line 170 and 180: It is a bit ambiguous to understand if the naming is either automatic or manual. Try to explain clearly and consider including an image solely dedicated to input data handling.

Processing of ROI

Lines 195-208: quite annoying to read through such a small amount of text calling back to so many supporting figures. Would it be possible to have a figure in main text or either organise the text differently?

Benchmarking of either PUPAID-, manual- or other state-of-the-art methods-generated

cell segmentation

Line 237: the section here seems a mix of user instructions and benchmarking. By the way, consider including a user interaction section along with UI images, would probably be very beneficial to promote the adoption of the tool.

Line 247: provide quantitative details about the test datasets

Line 252-253: what is the error on the single metric you are using to claim significance? There are hints in the images but this information is a bit tough to retrieve on the spot.

Results

Line 319: how is the past application of a novel tool a result? Consider providing a clear explanation. You could refer to validation on published studies, and simply include the reference

Line 342: Again this sounds like a validation rather than result section.

**Reviewer #2: **The manuscript PONE-D-24-13313 by P. Régnier and colleagues presents PUPAID, a workflow developed in R and ImageJ for semi-automated processing and analysis of multi-channel immunofluorescence data. They explain its workflow, which is user-friendly by incorporating an optional R Shiny-based interactive application for those not proficient in R. PUPAID performance is validated on few datasets and compared with state-of-the-art methods like StarDist or Cellpose, identifying conditions where it outperforms them, especially in dense areas. It exports single-cell data as FCS files, compatible with various cytometry analysis software. PUPAID is available as a GPLv3-licensed R package on GitHub. The work is well presented and with convincing experimental results and it is well written. The installation and usage is well documented and reproducible.

I support the publication of the manuscript without additional changes.

6. PLOS authors have the option to publish the peer review history of their article (what does this mean?). If published, this will include your full peer review and any attached files.

Reviewer #1: No

Reviewer #2: **Yes: **Vladislav Gavryusev

---

## [Author Response · Author response to Decision Letter 0]

27 Jul 2024

Response to Editor and Reviewers

Thank you for submitting your manuscript to PLOS ONE. After careful consideration, we feel that it has merit but does not fully meet PLOS ONE’s publication criteria as it currently stands. Therefore, we invite you to submit a revised version of the manuscript that addresses the points raised during the review process.

The revisions were positive, highlighting the manuscript's potential impact in research fields. However, while your work presents groundbreaking potential, its real advantages seem to be primarily in high-density regions. Given the clinical expertise among the authors, a detailed medical discussion would be highly beneficial.

From a reviewer's perspective, providing or at least describing the testing dataset is essential for comprehensive evaluation. Additionally, the manuscript lacks a thorough explanation of the benchmarking, which appears somewhat confusing. Improved communication on the quantitative methods used for benchmarking would significantly enhance the validation of PUPAID.

Journal Requirements:

 As requested by the Editor, we profoundly edited our manuscript to answer to the reviewers’ comments as well as to fully comply with the PLOS One journal requirements regarding the formatting of the manuscript. The revised version of the manuscript should fully embrace the journal requirements based on the two PDF resources the Editor provided to us.

As requested by the Editor, and given the provided URL regarding the recommended repositories, we uploaded the example dataset on the Data Station Life Sciences repository hosted by the Data Archiving and Networked Services (DANS). The associated accession URL (https://doi.org/10.17026/LS/7XQFAT) was indeed added within the revised version of the manuscript.

We thank the Editor for his remark. As requested in the previous point, we already uploaded the full example dataset on the Data Station Life Sciences repository hosted by the Data Archiving and Networked Services (DANS), which will be accessible via the following URL: https://doi.org/10.17026/LS/7XQFAT. Regarding the GitHub repository, all authors confirm that the code with be fully available if the manuscript is accepted for publication. Concretely, the developer and maintainer of PUPAID (Dr. Paul Régnier actually) only has to toggle a setting from “Private” to “Public” in his GitHub repository profile, which will instantly make the full code available to everyone. He again wanted to reassure the Editor that he will do this task without delay if the manuscript is accepted for publication. 

 We wanted to apologize for this point, as this was not intentional at all. The corresponding author probably copied and pasted within the online submission form a version of the abstract which was not the very last one. We carefully checked both abstracts and made sure they were identical between the manuscript and the online submission system.

Reviewers’s comments to the Author

Reviewer #1:

Thank you for your contribution with the development of PUPAID. While your work presents groundbreaking potential, according to this version of the manuscript the real advantages appear to be primarily in high-density regions. Given the clinical expertise available among the authors, it would be highly beneficial to leverage this by providing detailed medical discussion.

We thank the reviewer for his very important remark. To discuss this interesting point, we added a new Potential impact of PUPAID on the clinical and medical fields section in the manuscript which details how PUPAID could be useful in the clinical and medical fields. We notably expose here tissues and their associated diseases for which immunofluorescence is widely used for both diagnostic and research purposes, and for which standardized and robust analysis methods could potentially help a lot to better understand the diseases’ pathophysiology, but also to accelerate the discovery of new biomarkers and further refine the classification of patients.

From a reviewing point of view it would help to be provided with a testing dataset, or at least to describe it, for comprehensive evaluation. Additionally, the manuscript lacks a thorough explanation of the benchmarking which appears a bit confusing. I think venturing in many comparisons, different in nature, is not simple to deal with. Improved communication on the quantitative methods used for benchmarking would significantly enhance the validation of PUPAID.

We thank the reviewer for his remark and made modifications to improve the points he highlighted. First, to ease readers understanding, we transferred and renamed the Test dataset section present in the Supplemental Material and Methods to a new Example dataset for learning and test purposes subsection in the main manuscript in the Material and methods section in order to present the example dataset, and notably what kind of organ/patient is the tissue from, how it was stained (technology, antibodies and Opal reagents) and how it was acquired and processed (instrument, software and demultiplexing step). We also indicated the new URL from which users can download the dataset.

We additionally moved every supplemental methods from the Supplemental Material and Methods to the main manuscript. This lengthens the manuscript a bit but allows to avoid switching between files for an explanation of a precise point. In a similar manner, we decided to integrate the old Figure S1 as a new Figure 2 in the main manuscript, to directly provide to readers the totality of the benchmarking data.

 Second, we were unsure about what the reviewer specifically referred to when writing about the “benchmarking”, because the manuscript details two different ones. The first benchmarking was related to the CLAHE algorithm and helped to demonstrate that this step was pertinent and enhanced the subsequent cell segmentation through PUPAID. The second benchmarking was about the performance of PUPAID (with different σ ratios) as compared to manual cell segmentation but also to other well-established methods: StarDist and Cellpose. To improve the overall readability of the manuscript, we decided to edit both parts in the revised version of the manuscript, even if one was not directly targeted by the reviewer. In a sense, his remark tells us that the manuscript was probably not organized properly. To summarize, we transferred the Comparison of masks generated with different cell segmentation methods section originally present in the Supplemental Material and Methods to the main manuscript, in the Benchmarking of the different cell segmentation approaches subsection of the Material and methods section. We also extracted the CLAHE algorithm benchmarking part to a dedicated subsection between the Processing of ROI and Benchmarking of either PUPAID-, manual- or other state-of-the-art methods-generated cell segmentation subsections. This way, the methodology used for both benchmarking is totally visible and independent of the other sections, and the CLAHE-related section comes after the full description of the ROI processing. We believe this will improve the readability of the manuscript. These modifications indeed led to integrating the old Figure S1 as the new Figure 2, whereas the old Figure 2 is now the new Figure 3 and the old Figure 3 is now the new Figure 4.

I also think the paper lacks demonstration of user-friendliness, with the language used resulting in an additional barrier. Including a figure or, preferably, a video of the interface would greatly enhance understanding and usability for readers.

We greatly thank the reviewer for his very pertinent remark about the user-friendliness of PUPAID. Even if it was not as easy as one could think, we finally managed to generate a video of roughly 14 minutes which shows in live the full workflow of PUPAID applied on the example dataset. This video is accessible from the following YouTube link: https://youtu.be/58Tm54OVP-g.

Lastly, the manuscript's language is quite self-referential and would benefit from a more reader-centric approach, ensuring clarity and accessibility for a broader audience. I suggest giving more scientific context and sometimes guiding the reader via bullet points.

We greatly thank the reviewer for his constructive remark. To make our manuscript easier to read, we deeply modified its structure and content, as previously stated. First, the (now) full compliance with the PLOS One journal format helped a lot to restructure our work and our thoughts. Second, we rewrote entire sections of the manuscript to give more details (or less in some paragraphs) and to make a lot of sentences clearer. We also moved/switched full sentences/paragraphs to other (or even new) subsections to better structure the manuscript. Third, to enhance clarity and readability, we also put as main figures and methods the old supplemental materials originally submitted. This will ensure that readers will not have to permanently switch between documents when they read our manuscript. Fourth, we modified as much as we could the figures to make graphs and texts as big as possible to ease the reading. Fifth, we implemented a new subsection to add more scientific context regarding PUPAID’s utility and impact, notably in the medical field.

Minor changes:

Abstract

Line 58: Would be nice for reviewers to have test files to run the code

We thank the reviewer for his remark but wanted to clarify that test files (in this case, TIFF-formatted images containing multi-channel immunofluorescence data) were already available during the last submission of this manuscript, as mentioned in the last paragraph of the old Availability and future directions section of the originally submitted manuscript. If the reviewer encountered difficulty to access the files using the provided URL within the last weeks/months, we wanted to apologize for this issue, which was indeed not intentional. To address this problem in the long term, and in accordance with the Editor, we uploaded the test dataset to the Data Station Life Sciences repository hosted by the Data Archiving and Networked Services (DANS), where it should be available at all times. The test dataset will be totally accessible via the following URL: https://doi.org/10.17026/LS/7XQFAT.

Introduction

Line 82-83: would appreciate if the author could elaborate on what is meant by “exponential increase of the complexity”. Example: what is the role of noise and how is it evolving through time. Also it is not very clear what is an “early lack of open-source methods”

To be more precise, we wanted to underline the fact that over time, and especially during the two last decades, multiplex immunofluorescence underwent massive improvements together with a great diversification of methods, thus leading to the increase of the number of simultaneously acquired markers as well as the overall resolution and number of co-analyzed samples. We rephrased this sentence to be clearer and added references to emphasize on this point. We also removed the early lack of open-source methods part, because if it was true 10 to 20 years ago, it does not seem to be totally valid anymore, especially when we look at the plethora of analysis methods that were recently developed for this precise kind of purpose.

Line 88: The learning curve of coding within publicly available tools can be steep, conversely I would not define programs like Image J cumbersome to master.

We wanted to clarify what is said in this part of the manuscript. We did not mean to say that Image J is difficult to master. Actually, this software is very well programmed and organized, and offers the ability to quickly analyze heavy images with a broad variety of tools. However, if one needs to analyze dozens or hundreds of images using the same methodology in Image J, the need for automatization through macros and scripts (thus necessarily involving programming) is almost mandatory. To make this point of view clearer, we also rephrased this part.

Line 94: would be interesting if the author could elaborate on the relevance of Doc Parra in assessing methodology performances. Additionally, please provide further evidence of the status quo claimed here.

We thank the reviewer for his very constructive comment. Our initial sentence was meant to highlight the fact that the plethora of available analysis methods for immunofluorescence assays could potentially lead to the biased selection by researchers/scientists of methods that actually “produce” convenient results and thus avoid the “least performant” ones. We wanted to underline the notion that workflows which process data from A to Z (pre-processing, processing, normalization, analysis, results export, etc.) are one way to intent standardization of such analyses and avoid potential issues and biases. We thought this was relevant to write at first, but all your remarks taken as a whole finally lead us to doubt about the true relevance of this sentence in the introduction. In essence, our message is slightly suggested in the sentence the vast majority of the available methods only seems to propose a given kind of analysis and almost never include actual pre-processing and processing of the raw images written right before. In consequence, we thought it was better to remove the sentence related to Dr. Parra and thus rewrote the two last paragraphs of the introduction to make it clearer.

Lines 110-112: quite a statement about performance, please provide quantitative insights.

We thank the reviewer for his remark and added some quantitative insights in the last paragraph of the introduction but also added a new Table 2 which summarizes the efficacy of StarDist, Cellpose and PUPAID (using its most balanced 1:1.4 σ ratio and an IoU threshold of 0.4) independently for each measurement (% of detected objects and each of the four mentioned metrics) and in average. This will help readers to have a simpler “take home” message regarding PUPAID’s efficacy as compared to other well-established cell segmentation methods.

Dependencies

Lines 124-126: 

---

## [Decision Letter · Decision Letter 1]

5 Aug 2024

PUPAID: a R + ImageJ pipeline for thorough and semi-automated processing and analysis of multi-channel immunofluorescence data

PONE-D-24-13313R1

Dear Dr. Régnier,

We’re pleased to inform you that your manuscript has been judged scientifically suitable for publication and will be formally accepted for publication once it meets all outstanding technical requirements.

Kind regards,

Luca Pesce, Ph.D.

Academic Editor

PLOS ONE

Additional Editor Comments (optional):

Reviewers' comments:

Reviewer's Responses to Questions

**Comments to the Author**

1. If the authors have adequately addressed your comments raised in a previous round of review and you feel that this manuscript is now acceptable for publication, you may indicate that here to bypass the “Comments to the Author” section, enter your conflict of interest statement in the “Confidential to Editor” section, and submit your "Accept" recommendation.

Reviewer #1: All comments have been addressed

2. Is the manuscript technically sound, and do the data support the conclusions?

Reviewer #1: Yes

3. Has the statistical analysis been performed appropriately and rigorously? 

Reviewer #1: Yes

4. Have the authors made all data underlying the findings in their manuscript fully available?

Reviewer #1: Yes

5. Is the manuscript presented in an intelligible fashion and written in standard English?

Reviewer #1: Yes

6. Review Comments to the Author

Reviewer #1: Great job, really appreciated the hard work on making the manuscript more readable and sharing data from a more insightful perspective.

7. PLOS authors have the option to publish the peer review history of their article (what does this mean?). If published, this will include your full peer review and any attached files.

Reviewer #1: No

---

## [Editor Report · Acceptance letter]

11 Sep 2024

PONE-D-24-13313R1 

PLOS ONE

Dear Dr. Régnier, 

I'm pleased to inform you that your manuscript has been deemed suitable for publication in PLOS ONE. Congratulations! Your manuscript is now being handed over to our production team.

Kind regards, 

on behalf of

Dr. Luca Pesce 

Academic Editor

PLOS ONE